# A Modeling Study on Vaccination and Spread of SARS-CoV-2 Variants in Italy

**DOI:** 10.3390/vaccines9080915

**Published:** 2021-08-17

**Authors:** Chiara Antonini, Sara Calandrini, Fortunato Bianconi

**Affiliations:** 1ICT4life Srl, Via Mario Donati Guerrieri, 16, 06132 Perugia, Italy; sara.calandrini@i4l.company; 2Department of Engineering, University of Perugia, Via Goffredo Duranti, 93, 06125 Perugia, Italy; 3COVID-19 Epidemiological Unit, Regional Government of Umbria, Corso Vannucci, 96, 06121 Perugia, Italy; fortunato.bianconi@gmail.com

**Keywords:** SEIR model, COVID-19, SARS-CoV-2 variants, vaccines, model calibration, Italy

## Abstract

From the end of 2020, different vaccines against COVID-19 have been approved, offering a glimmer of hope and relief worldwide. However, in late 2020, new cases of severe acute respiratory syndrome coronavirus 2 (SARS-CoV-2) started to re-surge, worsened by the emergence of highly infectious variants. To study this scenario, we extend the Susceptible-Exposed-Infectious-Removed model with lockdown measures used in our previous work with the inclusion of new lineages and mass vaccination campaign. We estimate model parameters using the Bayesian method Conditional Robust Calibration in two case studies: Italy and the Umbria region, the Italian region being worse affected by the emergence of variants. We then use the model to explore the dynamics of COVID-19, given different vaccination paces and a policy of gradual reopening. Our findings confirm the higher reproduction number of Umbria and the increase of transmission parameters due to the presence of new variants. The results illustrate the importance of preserving population-wide interventions, especially during the beginning of vaccination. Finally, under the hypothesis of waning immunity, the predictions show that a seasonal vaccination with a constant rate would probably be necessary to control the epidemic.

## 1. Introduction

The outbreak of COVID-19, caused by the severe acute respiratory syndrome coronavirus 2 (SARS-CoV-2), has caused a global health and economic crisis, with over 167 million cases and 3 million deaths, at the time of writing [1].

Italy was the first European country strongly affected by the COVID-19 pandemic in early 2020. The first confirmed cases were detected at the end of February and, since then, the epidemic rapidly spread in the entire country, especially in the northern regions. From March 2020 until the beginning of May 2020, the Italian Government imposed a country-scale lockdown in order to reduce the spreading of the virus and mitigate the impact on the national health system [2]. Starting in May 2020, new positive cases and hospitalizations were drastically reduced and the intervention measures were progressively eased. However, the increased mobility and social gatherings caused a resurgence of SARS-CoV-2 infections at the end of August, leading the beginning of the second wave of the COVID-19 pandemic [2]. Despite the social distancing measures imposed, many factors, such as school reopening and crowded public transport, caused a new critical rise of infected cases and deaths, breaking the testing, tracing, and isolation program. This situation forced the Italian government to adopt a new color-coded system of restrictions, based on the risk scenario of each region [3]. The mitigation measures introduced, if compared to the first lockdown, were less restrictive, in order to be bearable from a social and economic point of view. As a result, the descending phase of the second wave was very slow and steady [4].

The second wave of infections was also characterized by two antithetical events, critical for the evolution of the pandemic: the discovery of new variants and the approval of the first vaccines. The novel lineages of SARS-CoV-2 were initially detected in Italy in December, and they rapidly gave rise to a third wave of infections in 2021 [5]. The so-called variants of concerns (VOC) are more transmissible and might be associated with higher disease severity than the original strain [6,7]. When the Italian government closed the borders to limit the introduction of new lineages, these variants were already spread across the country and they soon became dominant [8]. Moreover, while almost all regions experienced an uptick of the curve of contagion in March 2021, the Umbria region and the province of Bolzano were exposed to a high circulation of these lineages already between January and February, 2021 [9].

Since the first publishing of the genetic sequence of SARS-CoV-2 in January 2020, industry developers have struggled to develop a vaccine against this disease [10]. To date, there are four vaccines approved by the European Medicines Agency (EMA) and the Italian Medicines Agency (AIFA). The first to be approved are two mRNA vaccines, i.e., Pfizer/BioNTech and Moderna, which are administered in a two-dose regimen. They have 95% and 94.1% efficacy at preventing COVID-19 illness, respectively [11,12]. Then, two viral vector vaccines were also approved, the Vaxzevria vaccine by Oxford-AstraZeneca and the Johnson&Johnson’s vaccine. The Vaxzevria vaccine, administered in two doses 12 weeks apart, has an efficacy of 81.3%, while the Johnson&Johnson’s vaccine is one shot with an efficacy up to 85% [13,14]. The Italian vaccination campaign started on 27 December 2020 and was divided in four main phases [15]. Initially, the priority was given to medical workers together with guests and personnel from nursing homes. Then, the roll-out was organized according to age and clinical vulnerabilities of people.

Given this scenario, epidemiological models play a crucial role in exploring the future trajectory of COVID-19 and assessing the role of immunity response and the impact of new variants [16,17]. Various frameworks, based on the well known Susceptible-Infected-Recovered (SIR) model and its derivatives, have been developed to understand the nature of the pandemic and predict its possible evolution. In [18], the duration of the immune responses and the severity of secondary cases are studied through a Susceptible-Infected-Recovered-Susceptible (SIRS) model. In [6], an age-structured and regionally structured mathematical model is employed to understand the transmissibility of the B.1.1.7 lineage in United Kingdom (UK), while in [7], a Bayesian model analyzes the increase of infections due to the P.1 variant in Manaus, Brazil. As regards Italy, Giordano et al. evaluated the adoption of different restriction scenarios during the mass vaccination campaign with the compartmental model SIDARTHE-V [4].

Here, we extend the SEIR model with intervention parameters (SEIRL) presented in [19] to include the presence of new variants and vaccination roll-out (SEIRL-V). We focus on the case study of Italy and of the Umbria region, the first region displaying the emergence of variants. To faithfully reproduce the epidemic evolution, we estimate model parameters using our Bayesian method Conditional Robust Calibration (CRC) against the Italian and Umbria data [20,21,22]. By varying the vaccination rate and the relaxation of non-pharmaceutical interventions (NPIs), we simulate multiple scenarios to forecast the dynamics of COVID-19 in 2021 and beyond.

## 2. Materials and Methods

### 2.1. SEIRL-V Compartmental Model

In this study, we use and extend the SEIRL model presented in [19], with the introduction of vaccinations (SEIRL-V). The SEIRL-V compartmental model is shown in Figure 1 and it includes twelve infection states. Susceptible individuals (class *S*), when in contact with infected people, evolve into the exposed condition (class *E*) where they are infected but do not transmit the virus. Then, *E* individuals become pre-symptomatic (class $P_{S}$), meaning that they still do not have symptoms but are able to infect other people. After leaving the $P_{S}$ class, the group of infected is divided between asymptomatic (class *A*) and mild cases (class *M*). People in *M* class can progress to severe infection (class *H*), which requires hospitalization. Individuals in *H* may develop acute life-threatening symptoms and need treatment in an Intensive Care Unit (ICU) (class ICU). All infected classes can instead recover and evolve to the *R* class. Here, differently from the authors of [19], we suppose that both people in *H* and ICU class can die. Susceptible individuals are vaccinated with the injection of two doses (classes V1 and V2). After several days from the second dose, people acquire immunity (class Im). However, due to incomplete vaccine efficacy, vaccinated people may not develop the immunization and become infected, entering the *E* class. Even though vaccinated people contract the infection, they have a reduced probability of developing symptoms and do not need to be hospitalized. People in the Im and *R* classes have a finite period of immunity, after which they return to the susceptible class.

The model is a positive and bilinear system: all state variables have non-negative values for time t≥0, if initialized with non-negative values at time 0. Due to the rapid disease spread, human birth and death rates are omitted in the model [23]. For the mass conservation property, we have that $\dot{S}+\dot{E}+\dot{P_{S}}+\dot{A}+\dot{M}+\dot{H}+\dot{ICU}+\dot{R}+\dot{D}+\dot{V_{1}}+\dot{V_{2}}+\dot{Im}=0$, thus the sum of the states is constant and equal to the total population *N*.

The Ordinary Differential Equations (ODEs) system describing the interaction between these groups of population is as follows:(1)$$\begin{array}{cc} \; & \dot{S}=-\left(b_{e}P_{S,\nu }+b_{0}A+b_{1}M+b_{2}H+b_{3}ICU+\eta \right)S+\frac{R}{\delta }+\frac{Im}{\delta }\\ \; & \dot{E_{\nu }}=\left(b_{e}P_{S,\nu }+b_{0}A+b_{1}M+b_{2}H+b_{3}ICU\right)S_{\nu }-a_{0}E\\ \; & \dot{P_{S,\nu }}=a_{0}E_{\nu }-a_{1}P_{S,\nu }\\ \; & \dot{A}=z_{\nu }fa_{1}P_{S,\nu }-g_{0}A\\ \; & \dot{M}=\left(1-z_{\nu }\right)\left(1-f\right)a_{1}P_{S,\nu }-g_{1}M-p_{1}M\\ \; & \dot{H}=p_{1}M-g_{2}H-p_{2}H-u_{1}H\\ \; & \dot{ICU}=p_{2}H-g_{3}ICU-uICU\\ \; & \dot{R}=g_{0}A+g_{1}M+g_{2}H+g_{3}ICU-\frac{R}{\delta }\\ \; & \dot{D}=uICU+u_{1}H\\ \; & \dot{V_{1}}=\eta S-\frac{V_{1}}{\tau }-\left(b_{e}P_{S,\nu }+b_{0}A+b_{1}M+b_{2}H+b_{3}ICU\right)\left(1-\rho_{1}\right)V_{1}\\ \; & \dot{V_{2}}=\frac{V_{1}}{\tau }-\left(b_{e}P_{S,\nu }+b_{0}A+b_{1}M+b_{2}H+b_{3}ICU\right)\left(1-\rho_{2}\right)V_{2}-\frac{V_{2}}{\tau_{imm}}\\ \; & \dot{Im}=\frac{V_{2}}{\tau_{imm}}-\frac{Im}{\delta } \end{array}$$

The state variables of the ODEs are represented by the capital Latin letters:*S*: susceptible individuals,$E_{\nu }$: exposed individuals, where ν=0,1,2 denotes the numbers of vaccines doses received,$P_{S,\nu }$: pre-symptomatic individuals, where ν=0,1,2 denotes the numbers of vaccines doses received,*A*: asymptomatic individuals,*M*: people with mild infection,*H*: people in hospital with severe symptoms,ICU: people with critical infection which requires ICU level care,*R*: recovered individuals,*D*: dead people,$V_{1}$: people vaccinated with the first dose of vaccine,$V_{2}$: people vaccinated with the second dose of vaccine,Im: immune individuals.$S_{\nu }$ with ν=0,1,2 as the number of doses received. In more detail, $S_{0}=S$, $S_{1}=\left(1-\rho_{1}\right)V_{1}$ and $S_{2}=\left(1-\rho_{2}\right)V_{2}$.

Parameters $b_{i},\;\forall i=e,0,1,2,3$, are the transmission rates, representing a contact between a susceptible individual and an infected of classes $P_{S}$, *A*, *M*, *H*, and ICU, respectively. Parameters $g_{i}\;\forall i=0,1,2,3$ represent the different recovery rates of classes *A*, *M*, *H*, and ICU, respectively, while $u_{1}$ and *u* are the death rates of hospitalized and ICU people. Parameters $a_{i}\;\forall i=0,1$ indicate the rate of exit from classes *E* and $P_{S}$, while parameters $p_{1}$ and $p_{2}$ are the progression rate from mild to severe infection and from severe to critical infection. Parameter *f* is the fraction of asymptomatic individuals. All these model parameters are derived from the clinical observations using the following formulas:(1)$a_{1}=PresymPeriod^{-1}$(2)$a_{0}={\left(IncubPeriod-PresymPeriod\right)}^{-1}$(3)$g_{1}=DurMildInf^{-1}\cdot \left(1-FracSevere-FracCritical\right)$(4)$p_{1}=DurMildInf^{-1}-g_{1}$(5)$p_{2}=DurHosp^{-1}\cdot \frac{FracCritical}{\left(FracSevere+FracCritical\right)}$(6)$u_{1}=DurHosp^{-1}\cdot (\frac{\left(ProbDeathH\cdot \frac{FracSevere}{100}\right)}{FracSevere}$(7)$g_{2}=DurHosp^{-1}-p_{2}-u_{1}$(8)$u=TimeICUDeath^{-1}\cdot (\frac{\left(ProbDeath\cdot \frac{FracCritical}{100}\right)}{FracCritical}$(9)$g_{3}=TimeICUDeath^{-1}-u$(10)f=FracAsym(11)$g_{0}=DurAsym^{-1}$,
where PresymPeriod is the length of the infectious phase of incubation period IncubPeriod. DurAsym is the average duration of asymptomatic infection, DurMildInf is defined as the duration of mild symptoms or the time from symptom onset to hospitalization, for those who progress to class *H*. The duration of severe infection DurHosp is the time from hospital admission to recovery or death or ICU admission. TimeICUDeath is the time from ICU admission to recovery or death. All time duration are expressed in days (d). FracAsym is the percentage of infected people that develops asymptomatic infection while FracSevere and FracCritical are, respectively, the percentage of individuals requiring hospitalization and ICU-level care.

To describe the availability of an effective vaccine against SARS-CoV-2, we model the injection of two doses, as most of the approved vaccines adopt this regimen. All of them are characterized by different values of administration delays and efficacy. For the model, the value of vaccination parameters is chosen according to those of the Pfizer/BioNTech vaccine as it is currently the most administered in Italy [24]. The parameters related to vaccination are the following:parameter η is the rate of injection of the first dose. It is modeled as a piecewise constant function;parameter δ is the duration of natural and vaccinal immunity. We suppose that the average duration of natural and vaccinal immunity is equal to 8 months (240 days), according to [25,26];parameter τ is the time between the first and second dose of vaccine and it is set to 21 days [11];parameter $\tau_{imm}$ is the number of days between the second dose and the acquired immunity and it is set to 14 days [11];parameter $\rho_{1}$ is the efficacy of the first shot of vaccine and it is set to 0.8 [27];parameter $\rho_{2}$ is the efficacy of the second shot of vaccine and it is set to 0.95 [11];parameter $\rho_{3}$ is the efficacy of the first shot against hospitalization ad it is set to 0.808 [28,29];parameter $\rho_{4}$ is the efficacy of the second shot against hospitalization ad it is set to 0.946 [28,29];parameter $z_{\nu }$ with ν=0,1,2 is introduced to represent vaccine efficacy against disease. Thus, $z_{0}=0$, $z_{1}=\rho_{3}$ and $z_{2}=\rho_{4}$.

Regarding vaccine efficacy, we consider two forms of efficacy: protection against infection, represented by parameters $\rho_{1}$ and $\rho_{2}$, and efficacy against disease, through parameters $\rho_{3}$ and $\rho_{4}$. Indeed, as vaccines do not provide full protection (100%), breakthrough infections are still possible. Thus, people who have received all recommended doses of vaccine can still contract the infection. However, vaccinated people who get sick are likely to have milder symptoms and it is very rare for them to experience severe illness or die [16]. The Italian vaccination program is mainly based on the prioritization of the oldest age groups and of vulnerable people. As age classes are not included in our model, we take them into account through the introduction of a Hill function between hospitalized (*H*) and immunized people (Im), in order to model the dependent decline of hospital admissions. As a result, the ODE for class *H* becomes
(2)$$\dot{H}=p_{1}M-g_{2}H-p_{2}H-u_{1}H+\frac{1}{1+{\left(\frac{Im}{K}\right)}^{n}}u_{vax},$$
where *K* is the inverse feedback strength indicator and *n* is the Hill coefficient. Indeed, for a given *n*, increasing *K* reduces the repression level, and vice versa, while as *n* tends to infinity the Hill function resembles the step function [30]. The term $u_{vax}$ is introduced to take into account the start of the massive vaccination campaign:(3)$$u_{vax}=\left\{\begin{array}{cc} 0, & t<t_{vax}\\ 1, & t\geq t_{vax}, \end{array}\right.$$
where $t_{vax}$ is the first day of vaccination in Italy, i.e., 27 December 2020 [24].

The model includes also non-pharmaceutical interventions (NPIs) and easing of restrictions alternatively promoted by the Italian Government, depending on the epidemic data. All these measures are represented by parameter $s_{0}$. Thus, the transmission rates are multiplied as follows:(4)$$\begin{array}{c} b_{e,lock}=b_{e}\cdot s_{0}\\ b_{0,lock}=b_{0}\cdot s_{0} \end{array}$$

Parameters $b_{1}$, $b_{2}$, and $b_{3}$ are not scaled by any factor because we implicitly include the ban of detected positive people on leaving their houses and the employment of personal protective equipment (PPE) in hospitals, supposing that they are all requirements from the beginning of the second wave of the epidemic.

The presence of new variants is represented through the variation of parameters $b_{e}$ and $b_{0}$, as it has been stated that these variants are highly transmissible [6,7]. In order to introduce this hypothesis, the transmission rate parameters of presymtpomatic and asymptomatic infected are given by
(5)$$b_{e,lock}=\left\{\begin{array}{cc} b_{e,1}\cdot s_{0}, & t<t_{var}\\ b_{e,2}\cdot s_{0}, & t\geq t_{var} \end{array}\right.$$
(6)$$b_{0,lock}=\left\{\begin{array}{cc} b_{0,1}\cdot s_{0}, & t<t_{var}\\ b_{0,2}\cdot s_{0}, & t\geq t_{var} \end{array}\right.$$
where $t_{var}$ is the time of introduction of the new variants. Using the next generation matrix, the formula for computing the basic reproduction number $R_{0}$, i.e., the number of individuals infected by a single infected individual during his infectious period, is [4,19,31]:(7)$$R_{0}=N\left[\frac{b_{e}}{a_{1}}+f\frac{b_{0}}{g_{0}}+\left(1-f\right)\frac{1}{p_{1}+g_{1}}\left(b_{1}+\frac{p_{1}}{p_{2}+g_{2}+u_{1}}\left(b_{2}+b_{3}\frac{p_{2}}{u+g_{3}}\right)\right)\right].$$

The system admits a disease-free state equilibrium where the disease dies out and an endemic equilibrium [4,18]. It can be partitioned into three subsystems: the first one with susceptible individuals and individuals vaccinated with one or two doses of vaccine; the second one with all the infected; and the third one which includes healed, dead, and immunized. The system can be rewritten in feedback form with the infected subsystem as a positive linear subsystem subjected to a feedback signal *c*. Defining $x={\left[E_{\nu }\;P_{S,\nu }\;A\;M\;H\;ICU\right]}^{T}$, the subsystem is
(8)$$\dot{x}\left(t\right)=Fx\left(t\right)+dc\left(t\right)=\left[\begin{array}{cccccc} -a_{0} & 0 & 0 & 0 & 0 & 0\\ a_{0} & -a_{1} & 0 & 0 & 0 & 0\\ 0 & z_{\nu }fa_{1} & -g_{0} & 0 & 0 & 0\\ 0 & \left(1-z_{\nu }\right)\left(1-f\right)a_{1} & 0 & -(g_{1}+p_{1}) & 0 & 0\\ 0 & 0 & 0 & p_{1} & -(g_{2}+p_{2}+u_{1}) & 0\\ 0 & 0 & 0 & 0 & p_{2} & -(g_{3}+u) \end{array}\right]x\left(t\right)+\left[\begin{array}{c} 1\\ 0\\ 0\\ 0\\ 0\\ 0 \end{array}\right]c\left(t\right)$$
(9)$$c\left(t\right)=S_{\nu }\left(t\right)y_{S}\left(t\right)$$
(10)$$y_{S}\left(t\right)=\left[\begin{array}{cccccc} 0 & b_{e} & b_{0} & b_{1} & b_{2} & b_{3} \end{array}\right]x\left(t\right)$$
(11)$$y_{R}\left(t\right)=\left[\begin{array}{cccccc} 0 & 0 & g_{0} & g_{1} & g_{2} & g_{3} \end{array}\right]x\left(t\right)$$
(12)$$y_{D}\left(t\right)=\left[\begin{array}{cccccc} 0 & 0 & 0 & 0 & u_{1} & u \end{array}\right]x\left(t\right)$$
(13)$$\dot{S}\left(t\right)=-S\left(t\right)\left(y_{S}\left(t\right)-\eta \right)+\frac{1}{\delta }\left(R\left(t\right)+Im\left(t\right)\right)$$
(14)$$\dot{R}\left(t\right)=y_{R}\left(t\right)-\frac{R(t)}{\delta }$$
(15)$$\dot{D}\left(t\right)=y_{D}\left(t\right)$$
(16)$$\dot{V_{1}}\left(t\right)=\eta S\left(t\right)-\frac{V_{1}\left(t\right)}{\tau }-\left(1-\rho_{1}\right)y_{S}\left(t\right)V_{1}\left(t\right)$$
(17)$$\dot{V_{2}}\left(t\right)=\frac{V_{1}\left(t\right)}{\tau }-\frac{V_{2}\left(t\right)}{\tau_{imm}}-\left(1-\rho_{2}\right)y_{S}\left(t\right)V_{2}\left(t\right)$$
(18)$$\dot{Im}\left(t\right)=\frac{V_{2}\left(t\right)}{\tau_{imm}}-\frac{Im(t)}{\delta }$$

### 2.2. Data

We calibrate the model against hospitalized, ICU, and dead patients (classes *H*, ICU, and *D* in the model, respectively). Epidemiological data about the COVID-19 evolution are available in the GitHub repository of the Italian Civil Protection Department [32]. Data about total number of vaccines administered are taken from the Github repository of the Italian Government [24].

### 2.3. Conditional Robust Calibration (CRC) for Parameter Estimation

CRC is an iterative algorithm for parameter estimation, belonging to the class of Approximate Bayesian Computation Sequential Monte Carlo (ABC-SMC) methodologies. A detailed description of CRC can be found in [19,20,21,22]. CRC is based on the sampling of the parameter space and it considers the parameter vector as a random variable **P**. It returns in output an approximation of the parameter posterior distribution conditioned to the available data $f_{P|y^{*}}\left(p\right)$, where $y^{*}$ is the dataset. At each iteration, CRC generates a matrix $P_{O}$ of $N_{S}$ parameter vectors through Latin Hypercube Sampling (LHS). Each parameter vector p is sampled inside an interval between a lower and upper boundary, $L^{1}$ and $U^{1}$, chosen by the user. Parameters are assumed to be uniformly or logarithmically distributed in their intervals. Then, for each sample $p\in P_{O}$, the ODE model is integrated to compute the *in silico* vector of observables **y**. The fitting between the simulated vector **y** and the dataset $y^{*}$ is measured through the Absolute Distance Function (ADF):(19)$$ADF_{i}=\sum_{j=1}^{k}\left|y_{i}\left(t_{j}\right)-y_{ij}^{*}\right|\;i=1,...,m,$$
where $y_{i}$ is the simulated observable at time point $t_{j}$, and $y_{ij}^{*}$ is the corresponding measured variable. This distance function measures the distribution of the error between simulated and real data when the parameters are sampled in each iteration. For each **p** $\in P_{O}$, each $ADF_{i}\;i=1,...,m$ is computed, and we select only those distance functions under a user defined threshold $\epsilon_{i}\geq 0$. Thus, we obtain different parameter sets $P_{O,\epsilon_{i}}$, one for each output variable. Each set contains only those parameters that yield the values of a specific distance function under the corresponding threshold. All these sets are intersected to obtain $P_{O,\epsilon }=\left\{{⋂}_{i=1}^{m}P_{O,\epsilon_{i}}\right\}$, where $\epsilon =\{\epsilon_{1},...,\epsilon_{i},...,\epsilon_{m}\}$. Using a kernel density approach, the approximate posterior distribution $f_{P|P_{O,\epsilon }}$ is estimated. This procedure is repeated for multiple iterations, updating the sampling interval on the basis of the posterior distribution of the previous iteration. The final output of the algorithm is $f_{P|P_{O,\epsilon }}$, where ϵ is the set of thresholds chosen in the final CRC iteration. The code for running CRC and the SEIRL-V model of COVID-19 is available at https://github.com/fortunatobianconi/CRC (accessed on 7 June 2021).

## 3. Results

### 3.1. Spread of Sars-CoV-2 Lineages in Umbria and Italy

During the second and third wave of COVID-19, several novel variants of Sars-CoV-2 emerged in Italy. Some of these lineages, i.e., B.1.1.7, B.1.351, and P.1, are known as VOC, as they may be characterized by increasing transmissibility, more severe disease and reduced cross-protective immunity [33]. The B.1.1.7 lineage was first detected in the UK in September 2020 and then it rapidly spread in at least 114 countries [6]. Around mid-November 2020, the P.1 strain emerged in Manaus, Brazil, causing a rapid resurgence of Sars-CoV-2 hospitalizations and new positive cases [7]. Another fast-spreading variant is the B.1.351, first reported in South Africa in December 2020 [34]. The proportion of COVID-19 cases imputable to these VOC has rapidly increased worldwide, replacing previously circulating variants. In Italy, the presence of variants was initially reported in late 2020 and the outbreak progressed to a peak in late March, when more than 20,000 new cases were notified every day. The B.1.1.7 lineage was the first one identified and, according to the Italian National Institute of Health (ISS), it became the prevalent one in a few months. Indeed, we started from a prevalence of 17.8% at the beginning of February to a prevalence of 91.6% at the end of April 2021 [8,35]. However, in the Umbria region, especially in the province of Perugia, the new variants pushed up the contagion curve way before the other Italian regions. Umbria was classified by the ISS as the epicenter of the P.1 lineage in Italy, with a prevalence of 36.2% in mid-February [36]. Indeed, as shown in Figure 2, the Umbria region experienced a peak of hospitalizations in February rather than in March, with a number of ICU admissions almost twice as much the one in Italy.

### 3.2. Umbria Case Study

The SEIRL-V model is calibrated using the CRC algorithm against data of Umbria from 1 September 2020 to 1 May 2021. We normalize data over the population N=882,000 and multiply them by $10^{5}$. According to the work in [32], initial conditions are set as follows: $S_{0}=10^{5}-E_{0}$, $E_{0}=\left(400/N\right)\times 10^{5}$, $P_{S,0}=300$, $A_{0}=144$, $M_{0}=144$, $H_{0}=8$, $ICU_{0}=2$, $R_{0}=1452$, $D_{0}=80$, $V_{1,0}=0$, $V_{2,0}=0$, $Im_{0}=0$. From September, many measures were issued by the national and local governments in order to contain the contagion curve. In the model, we consider the most relevant ones:(1)14 September 2020 ($T_{lock,1}$), school reopening;(2)19 October 2020 ($T_{lock,2}$), the Regional Government adopted some preventative measures such as remote teaching for part of the students, limited capacity of public transportation and closure of shopping malls during the weekend [37];(3)11 November 2020 ($T_{lock,3}$), Umbria region is classified as “orange”, i.e., as a medium-risk contagion zone;(4)6 December 2020 ($T_{lock,4}$), Umbria goes back to “yellow” zone, i.e., with moderate risk of virus spread;(5)7 January 2021 ($T_{lock,5}$), school reopening and easing of some restrictions after the country-wide red area;(6)8 February 2021 ($T_{lock,6}$), “red” area for the entire Province of Perugia, i.e., the highest level of restrictions, following an improvement in the contagion data and the identification of variants;(7)22 March 2021 ($T_{lock,7}$), back to ’orange’ zone with reopening of schools for the youngest.

Thus, the parameter vector that reproduces all these measures is $s_{0}=[s_{01},s_{02},s_{03},s_{04},$
$s_{05},s_{06},s_{07}]$, which modifies the transmission rate parameters as follows:(20)$$b_{e,lock}=\left\{\begin{array}{cc} b_{e}, & t<T_{lock,1}\\ b_{e}\cdot s_{0,i}, & T_{lock,i}\leq t<T_{lock,i+1}\;i=1,...,6\\ b_{e}\cdot s_{0,7} & t\geq T_{lock,7} \end{array}\right.$$
(21)$$b_{0,lock}=\left\{\begin{array}{cc} b_{0}, & t<T_{lock,1}\\ b_{0}\cdot s_{0,i}, & T_{lock,i}\leq t<T_{lock,i+1}\;i=1,...,6\\ b_{0}\cdot s_{0,7} & t\geq T_{lock,7} \end{array}\right.$$

Parameters $b_{e}$ and $b_{0}$ are differentiated also to account for the presence of new virus variants, starting from 19 December 2020 ($t_{var}$), around a month before the resurgence of new infections. Moreover, to accurately simulate the epidemic in Umbria, we vary the fraction of critical infected patients FracCritical as follows:$FracCritical=FracCritical_{1}$ from day 0 (1 September 2020) to day 35 (5 October 2020);$FracCritical=FracCritical_{2}$ from day 36 to day 83 (22 November 2020);$FracCritical=FracCritical_{3}$ from day 84 to day 152 (30 January 2021);$FracCritical=FracCritical_{4}$ from day 153 to day 200 (19 March 2021);$FracCritical=FracCritical_{5}$ from day 201 onward.

Finally, the vaccination rate η is a piecewise constant and is set in order to resemble the gradually increasing trend of first doses injected, as shown in Figure 3a. Thus, $\eta =[0,5.22\times 10^{-4},7.97\times 10^{-4},0.0027,0.0041]$ at days [0,118,154,181,212], as depicted in Figure 3b.

The parameter vector to estimate for Umbria contains twenty-four model parameters and seven interventions parameters, i.e., $p\in R^{31}$. As regards CRC, we set $N_{S}=10^{5}$ the number of samples in the parameter space, and we perform 10 iterations of the method. We repeat each iteration for 10 times, in order to ensure reliability of results. Table 1 shows the prior distribution chosen at the beginning of the calibration and the mode of the approximate posterior distribution computed by CRC in the 10th iteration for one of the realizations. The initial range of transmission rates is taken from [19]. For the intervention parameters, we suppose a range of variation between 0.1 and 0.9 if the corresponding event is supposed to curb the virus spread while we suppose a range between 0.4 and 1.5 if the associated event might foster the resurgence of new cases.

According to Table 1, CRC estimates that the spread of new SARS-CoV-2 lineages caused a percentage increase of 50.6% for the pre-symptomatic transmission rate and of 15.5% for the asymptomatic transmission rate. Consequently, the basic reproduction number $R_{0}$ is evaluated equal to 2.28 at the beginning of September and then goes up to 2.75 at the end of December. Figure 4 depicts the estimation of the time evolution for H, ICU and D variables, in comparison with the data.

Then, we simulate the evolution of the epidemic in the summer by comparing different vaccination schedules and the progressive loosening of restrictions, as shown in Table 2. We hypothesize three vaccine roll-outs, a slow one with $8\times 10^{3}$ first doses of vaccine every day, a medium one with $12.5\times 10^{3}$, and a fast one with $18\times 10^{3}$ per day. In parallel, we reduce restrictions, supposing a progressive reopening, divided in three main steps, according to the work in [38]:26 April 2021: reintroduction of the low-risk “yellow” zone;24 May 2021: curfew extension, gym reopening and restaurants with indoor seating;21 June 2021: curfew lifted and holiday season.

As before, we assume an intervention strategy with a mild, moderate, or high impact on the transmission parameters.

Figure 5, Figure 6 and Figure 7 show the evolution of the epidemic dynamics in all these nine possible scenarios until October 2021. Finally, Figure 8 reports the area plots for four scenarios chosen among the nine simulated, over a two-year time period in order to evaluate the long-term effects of vaccination and waning immunity. The four scenarios selected are Low/Low, High/Low, Low/Low, and High/High (see Table 2). In all cases, we evaluate the achievement of herd immunity, i.e., when the majority of the population is immune to the disease and the rate of infections goes down without restrictions. We define the 70% of the population as the herd immunity threshold, according to [39,40]. We consider to be immune from COVID-19 recovered people (class *R*), immune people (class Im) and people who have received the second dose of vaccine ($V_{2}$) as, at least in Italy, the so-called COVID-19 Green Pass is released 15 days after the first vaccine jab [41]. For Umbria, the model predicts that herd immunity is reached between the end of July and the beginning of August 2021 when $18\times 10^{3}$ vaccine doses are administered per day (Low/High and High/High scenarios). On the other hand, with $8\times 10^{3}$ vaccines per day, it is achieved at the beginning of September 2021 in the High/Low scenario while, in the Low/Low case, the percentage of immune people is stabilized at about 67%. In all cases, the number of hospitalizations becomes very small, without overwhelming the health care system.

### 3.3. Italy Case Study

As regards Italy, the model is calibrated from 1 September 2020 to 1 May 2021. According to the work in [32], initial conditions are $S_{0}=10^{5}-E_{0}$, $E_{0}=\left(30,000/N\right)\times 10^{5}$, $P_{S,0}=20,000$, $A_{0}=15,000$, $M_{0}=26,271$, $H_{0}=1437$, $ICU_{0}=109$, $R_{0}=208,201$, $D_{0}=35,497$, $V_{1,0}=0$, $V_{2,0}=0$, $Im_{0}=0$. Data are normalized over the Italian population $N=60\times 10^{6}$ and multiplied by $10^{5}$. From the beginning of September 2020, the Italian Government implemented multiple containment measures, more and more restrictive as infections soar. In November, Italian regions were divided in three risk zones on the basis of contagion data. There are high-risk ’red’ zones where only essential movements are allowed. In medium-risk ’orange’ zones, movements are allowed only in the same municipality and bars and restaurants can do only takeaway. In moderate-risk ’yellow’ zones, restrictions are less stringent as, for example, all shops can be open and restaurants can serve people outdoors. Moreover, a nationwide curfew from 10 pm to 5 am was approved to limit night movements. A detailed description of all the restrictions adopted can be found in [42].

Since multiple containment measures were implemented by the Italian Government from the beginning of September 2021, we consider the most significant ones in the model:(1)14 September 2020 ($T_{lock,1}$), school reopening;(2)6 November 2020 ($T_{lock,2}$), introduction of a three-tier color coded system of restrictive measures, based on the risk profile of each region;(3)24 December 2020 ($T_{lock,3}$), country-wide lockdown for Christmas holidays;(4)7 January 2021 ($T_{lock,4}$), school reopening and easing of some restrictions after the country-wide red area;(5)15 March 2021 ($T_{lock,5}$), removal of ’yellow’ zone in the color-coded system, leaving only medium and high risk zones.

Thus, the parameter vector $s_{0}=\left[s_{01},s_{02},s_{03},s_{04},s_{05}\right]$ changes the transmission rate parameters in the following way:(22)$$b_{e,lock}=\left\{\begin{array}{cc} b_{e}, & t<T_{lock,1}\\ b_{e}\cdot s_{0,i}, & T_{lock,i}\leq t<T_{lock,i+1}\;i=1,...,4\\ b_{e}\cdot s_{0,5} & t\geq T_{lock,5} \end{array}\right.$$
(23)$$b_{0,lock}=\left\{\begin{array}{cc} b_{0}, & t<T_{lock,1}\\ b_{0}\cdot s_{0,i}, & T_{lock,i}\leq t<T_{lock,i+1}\;i=1,...,4\\ b_{0}\cdot s_{0,5} & t\geq T_{lock,5} \end{array}\right.$$

As, in most of the Italian regions, the emergence of new lineages determined a growth of new hospitalization and deaths about a month later than Umbria (see Figure 2), we update parameters $b_{e}$ and $b_{0}$ on 19 January 2021 ($t_{var}$). The fraction of patients in ICU is varied as follows: $FracCritical=FracCritical_{1}$ from day 0 (1 September 2020) to day 35 (5 October 2020), $FracCritical=FracCritical_{2}$ from day 36 to day 77 (16 November 2020), $FracCritical=FracCritical_{3}$ from day 78 to day 117 (26 December 2020), $FracCritical=FracCritical_{4}$ from day 118 onward. The vaccination rate η is set as $\eta =[0,5.9\times 10^{-4},0.0018,0.0031,0.0044]$ at days [0,118,168,205,221], as depicted in Figure 9.

The parameter vector to estimate contains twenty-three model parameters and five interventions parameters, i.e., $p\in R^{28}$. Tuning parameters of CRC are set in the same way as Umbria, except for the number of iterations that is set to 11. The result of the calibration for H, ICU, and D state variables is shown in Figure 10. As for Umbria, CRC estimates a variation of the transmission rate parameters due to the new variants. The pre-symptomatic rate is characterized by a percentage increase of 87.3% while the asymptomatic rate undergoes a minimal decrease of 9.9%. Given these estimates, $R_{0}$ varies from an initial value of 1.94 in September to a value of 2.17 in January. As regards the other parameters, most of them are estimated with similar values for both Umbria and Italy, such as the pre-symptomatic period and the transmission rates of mild and hospitalized patients (see Table 1).

Finally, Figure 11, Figure 12 and Figure 13 depict the dynamics of the epidemic in the nine scenarios presented in Table 2. For Italy, the slow vaccination schedule is set equal to $5\times 10^{5}$ first doses of vaccine every day, the medium one is $8\times 10^{5}$, and the fast one $10^{6}$ per day. The intervention measures are implemented in the same way of Umbria. Figure 14 reports the area plots for the same four scenarios of Umbria, over a two-year time period. In case of fast vaccination, herd immunity is achieved at the end of August 2021 while in case of slow vaccination, the number of immune people is stable at 65% but with a negligible number of hospitalization and deaths.

As regards the computational cost, CRC takes around 20 minutes to complete one iteration. All the simulations are performed using Matlab (R2019a) on a Intel Core i7-4700HQ CPU, 2.40 GHz 8, 16-GB memory, Ubuntu 18.04 LTS (64 bit).

## 4. Discussion

This study highlights how the control of COVID-19 pandemic can be accomplished by a multi-strategy approach that combines high efficacy vaccines, social distancing, and isolation of detected cases.

Through model calibration with CRC, we are able to reproduce the second and third wave of the epidemic and understand the complex lineage turnover of COVID-19 in both presented case studies. First of all, CRC estimates similar values for the pre-symptomatic transmission rate $b_{e}$ in Italy and Umbria during the second wave, while the asymptomatic transmission rate $b_{0}$ is much higher in Umbria. These values explain the major incidence of new cases in Umbria during October and early November 2020. Indeed, at the end of October, the incidence over $10^{5}$ inhabitants was 458.49 in Umbria and 279.72 in Italy [43,44]. As shown in Figure 2, also the number of patients in ICU in Umbria was over the median national range. The restrictive measures adopted between November and December helped in the containment of this critical scenario at a regional and national level, until the emergence of the new VOC. Our model confirms the high transmissibility of the B.1.1.7 and P.1 lineages, through the increase of transmission rate parameters. However, while CRC estimates a growth of parameter $b_{e}$ in both case studies, parameter $b_{0}$ is estimated to rise only in Umbria, between the second and third wave. This is due to the stronger circulation of the P.1 variant from the beginning of 2020, as stated also by the ISS [35]. This context, considering that the vaccination campaign was only at the beginning, has led to a rapid resurgence of cases and made Umbria one of the worst-affected regions in the country during the third wave. The ’red’ area imposed in the province of Perugia in February reduced virus transmission by 70% (parameter $s_{06}$). The removal of ’yellow’ zone and the prohibition of travel between regions has caused a decrease in transmission by 80% (parameter $s_{05}$) in the entire country. Thus, containment measures were effective, especially in reducing the spread of the P.1 variant from Umbria to the other regions.

We then simulate different scenarios to evaluate the effects of mass vaccination campaign at different rates, combined with waning immunity and gradual reopening. In most predictions, the epidemic diffusion is contained and hospitalizations are reduced in both Umbria and Italy. The gap between slow and fast vaccination is not particularly significant if the reopening policy is low or medium. Things change when an high strategy of reopening is adopted. At a medium and high vaccination speed, there is a minimum increase of H and ICU variable at the end of summer 2021. On the other hand, at a slow vaccination rate, the growth in hospitalization may be more substantial but still under the peak of the third wave. This is probably due to the reverse age order of the vaccination schedule, which has immunized most of the elderly and vulnerable people. Moreover, because of the presence of variants, in Umbria the rise of hospital admissions may cause new pressure on the regional healthcare system. Thus, it is important to highlight that during the first part of the mass vaccination roll-out, the use of NPIs is pivotal to avoid the circulation of variants and the emergence of possible immune escape lineages.

Over longer time scales, model predictions reveal that waning immunity and vaccination play a crucial role in the eradication of SARS-CoV-2. Seasonal vaccination at a constant rate will be probably required, in order to control the epidemic diffusion and prevent a new wave of hospitalizations. COVID-19 will most likely become a manageable infection that will continue to circulate but with smaller and rarer outbreaks. However, we still lack precise information on the duration of immunity and on the severity of second infections [16,45]. Moreover, the value of herd immunity threshold is still under study as it depends on population heterogeneity and virus transmission dynamics [46,47].

Finally, this study contains some limitations. Our SEIRL-V model is a compartmental model which does not take into account age classes and seasonality in the transmission rates, as it is common for most respiratory diseases. Furthermore, it would also be interesting to study how long term predictions are affected by the variation of vaccine and natural immunity.

## 5. Conclusions

Our work presents a new SEIRL-V model for exploring multiple epidemiological scenarios in the presence of new SARS-CoV-2 variants and of high-efficacy vaccines. The calibrated model reproduces the high transmissibility of new virus variants and underlines the need of maintaining social distancing and case isolation, particularly during the first phase of the vaccination campaign. Then, it shows how effective vaccines are crucial in the long-term control of COVID-19.

## Figures and Tables

**Figure 1 vaccines-09-00915-f001:**
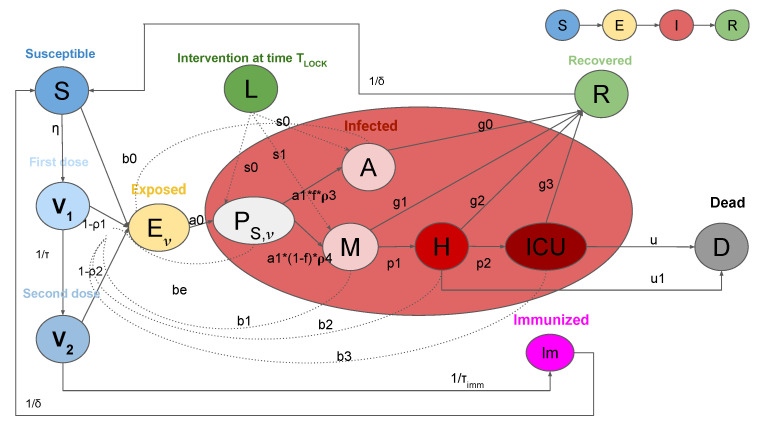
Graphic representation of the SEIRL-V model. Clinical stages for the population are as follows: Susceptible (*S*), Exposed ($E_{\nu }$), Presymptomatic ($P_{S,\nu }$), Asymptomatic (*A*), Recovered (*R*), Mild infection (*M*), Severe infection (*H*), Critical infection (ICU), Dead (*D*), Vaccinated 1st dose ($V_{1})$, Vaccinated 2nd dose ($V_{2})$, and Immunized (Im). The intervention measures are represented by *L*.

**Figure 2 vaccines-09-00915-f002:**
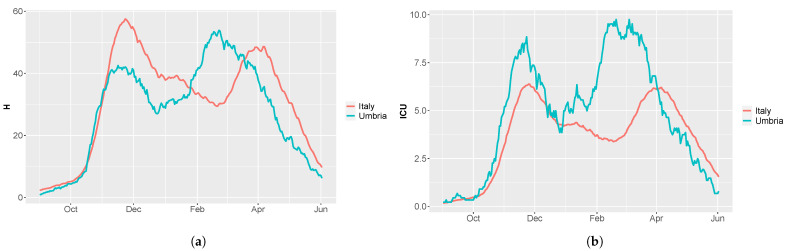
Comparison of hospitalization and ICU admissions in Umbria (light blue line) and Italy (red line) from 1 September until 1 May 2021. (**a**) Daily hospitalizations normalized over the whole population (∼882,000 for Umbria and ∼60 million for Italy) and multiplied by $10^{5}$. (**b**) Daily ICU admissions normalized over the whole population and multiplied by $10^{5}$.

**Figure 3 vaccines-09-00915-f003:**
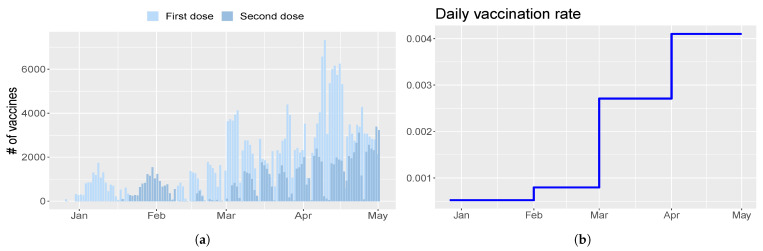
Vaccination campaign in Umbria. (**a**) Number of first (light blue) and second (blue) doses injected every day. (**b**) Profile of the evolution of the vaccination rate η chosen for model calibration.

**Figure 4 vaccines-09-00915-f004:**
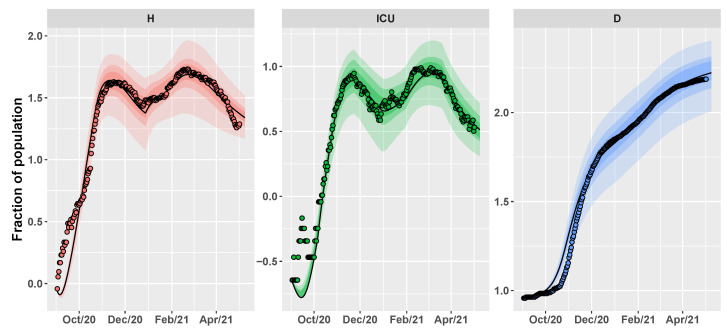
Umbria. Time behavior of H, ICU and D variables using as parameter vector the final mode vector computed by CRC (black line) (see Table 1); dots are the public data available at [32]. Both data and simulations are in log-scale, normalized over the population of Umbria (∼882,000) and multiplied by $10^{5}$. The colored area represents the variation of the temporal behavior when the parameter vector varies between the 60th, 70th, and 90th percentile of its conditional probability density function (pdf) (see Table A1).

**Figure 5 vaccines-09-00915-f005:**
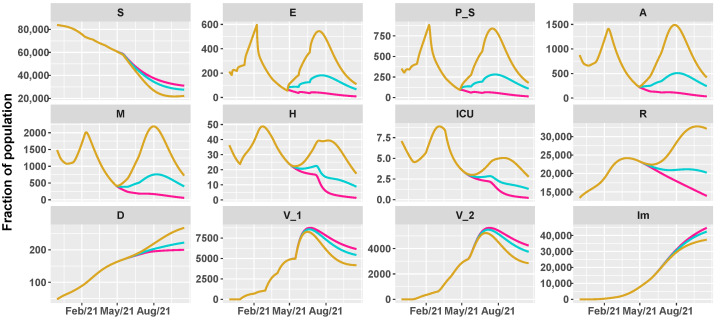
Projections of the epidemic evolution with a vaccination schedule of 8000 first doses of vaccines per day in Umbria (slow vaccination). Each line corresponds to a different reopening strategy, i.e., different values of transmission rate parameters $b_{e}$ and $b_{0}$ (yellow for high values, blue for medium values, and pink for low values).

**Figure 6 vaccines-09-00915-f006:**
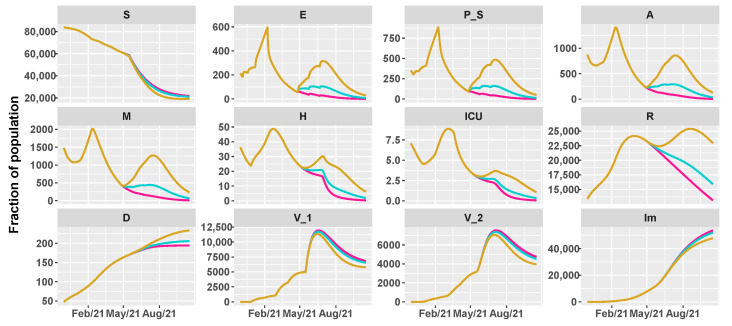
Projections of the epidemic evolution with a vaccination schedule of 12,500 first doses of vaccines per day in Umbria (medium vaccination). Each line corresponds to a different reopening strategy, i.e., different values of transmission rate parameters $b_{e}$ and $b_{0}$ (yellow for high values, blue for medium values, and pink for low values).

**Figure 7 vaccines-09-00915-f007:**
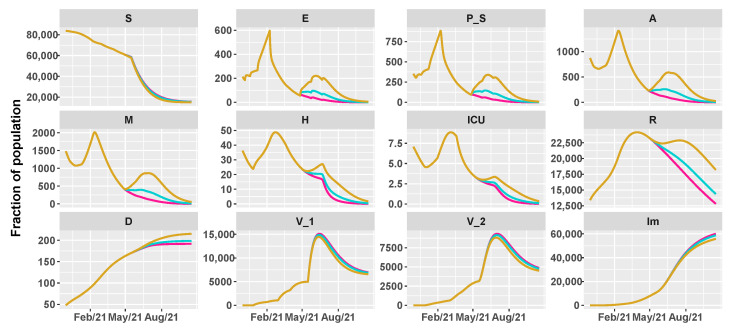
Projections of the epidemic evolution with a vaccination schedule of 18,000 first doses of vaccines per day in Umbria (fast vaccination). Each line corresponds to a different reopening strategy, i.e., different values of transmission rate parameters $b_{e}$ and $b_{0}$ (yellow for high values, blue for medium values, and pink for low values).

**Figure 8 vaccines-09-00915-f008:**
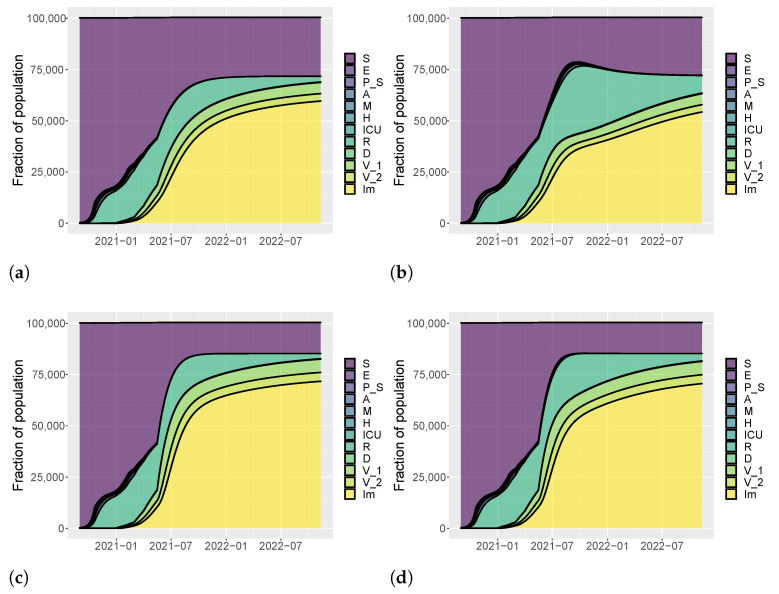
Area plots of model variables to compare the long-term evolution of the epidemic dynamics in Umbria. (**a**) 8000 first doses every day and low values for $b_{e}$ and $b_{0}$ (Low/Low). (**b**) 8000 first doses every day and high values for $b_{e}$ and $b_{0}$ (High/Low). (**c**) 18,000 first doses every day and low values for $b_{e}$ and $b_{0}$ (High/Low). (**d**) 18,000 first doses every day and high values for $b_{e}$ and $b_{0}$ (High/High).

**Figure 9 vaccines-09-00915-f009:**
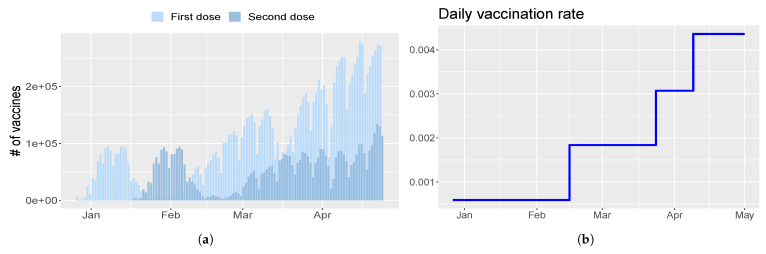
Vaccination campaign in Italy. (**a**) Number of first (light blue) and second (blue) doses injected every day. (**b**) Profile of the evolution of the vaccination rate η chosen for model calibration.

**Figure 10 vaccines-09-00915-f010:**
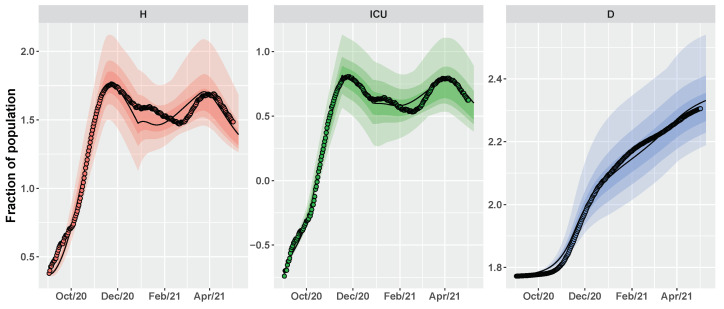
Italy. Time behavior of H, ICU, and D variables using as parameter vector the final mode vector computed by CRC (black line) (see Table 1); dots are the public data available in [32]. Both data and simulations are in log-scale, normalized over the population of Italy (∼60 million) and multiplied by 100,000. The colored area represents the variation of the temporal behavior when the parameter vector varies between the 60th, 70th and 90th percentile of its conditional pdf (see Table A2).

**Figure 11 vaccines-09-00915-f011:**
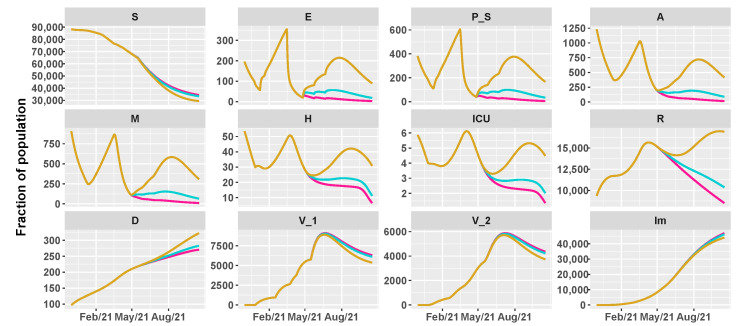
Projections of the epidemic evolution with a vaccination schedule of 500,000 first doses of vaccines per day in Italy (slow vaccination). Each line corresponds to a different reopening strategy, i.e., different values of transmission rate parameters $b_{e}$ and $b_{0}$ (yellow for high values, blue for medium values, and pink for low values).

**Figure 12 vaccines-09-00915-f012:**
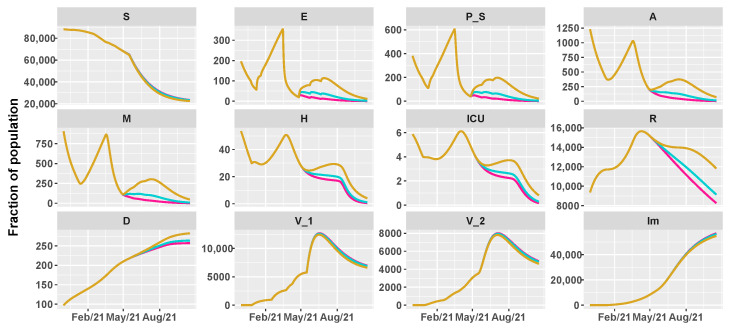
Projections of the epidemic evolution with a vaccination schedule of 800,000 first doses of vaccines per day in Italy (medium vaccination). Each line corresponds to a different reopening strategy, i.e., different values of transmission rate parameters $b_{e}$ and $b_{0}$ (yellow for high values, blue for medium values, and pink for low values).

**Figure 13 vaccines-09-00915-f013:**
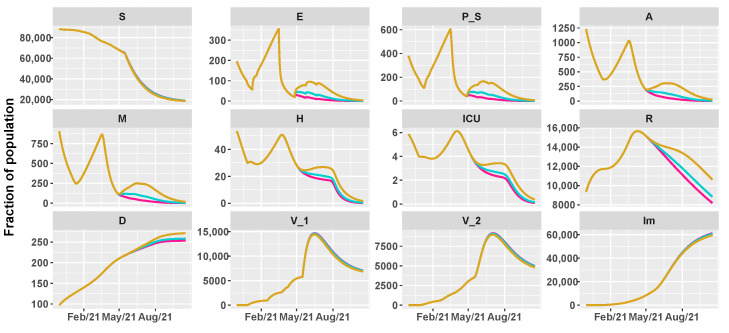
Projections of the epidemic evolution with a vaccination schedule of 1,000,000 first doses of vaccines per day in Italy (fast vaccination). Each line corresponds to a different reopening strategy, i.e., different values of transmission rate parameters $b_{e}$ and $b_{0}$ (yellow for high values, blue for medium values, and pink for low values).

**Figure 14 vaccines-09-00915-f014:**
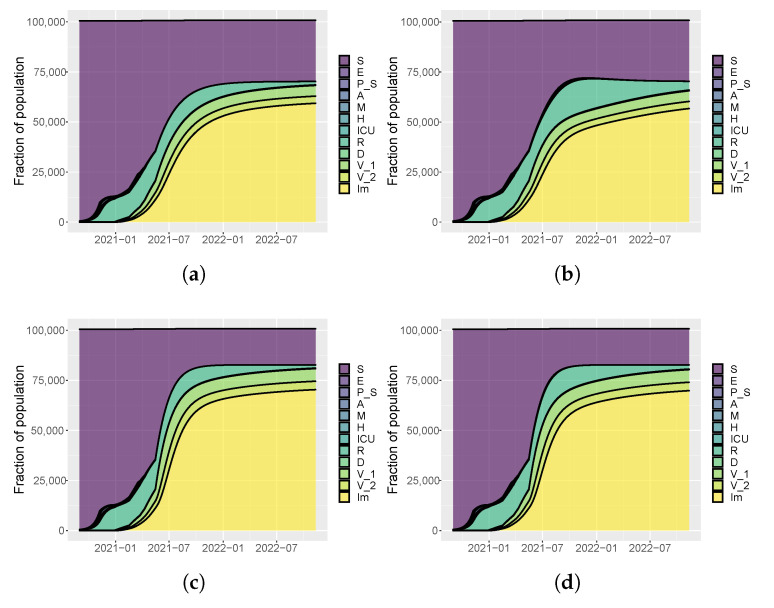
Area plots of model variables to compare the long-term evolution of the epidemic dynamics in Italy. (**a**) 500,000 first doses every day and low values for $b_{e}$ and $b_{0}$ (Low/Low). (**b**) 500,000 first doses every day and high values for $b_{e}$ and $b_{0}$ (Low/High). (**c**) 1,000,000 first doses every day and low values for $b_{e}$ and $b_{0}$ (High/Low). (**d**) 1,000,000 first doses every day and high values for $b_{e}$ and $b_{0}$ (High/High).

**Table 1 vaccines-09-00915-t001:** CRC results for Umbria and Italy. The second column shows the prior distribution of parameters, set at the beginning of the first CRC iteration. The third and fourth columns show, respectively, the mode vector of $f_{P|P_{O,\epsilon }}$ in one of the 10 final realizations for Umbria and Italy. Note that the pre-symptomatic period (PresymPeriod) is supposed to be a percentage of the incubation period (IncubPeriod).

Parameter	Prior	Umbria $p_{\mathrm{mode}}$	Italy $p_{\mathrm{mode}}$
$b_{e,1}$	log-U(0.01,1)	0.1442	0.1342
$b_{0,1}$	log-U(0.01,1)	0.3178	0.2109
$b_{e,2}$	log-U(0.01,1)	0.2172	0.2512
$b_{0,2}$	log-U(0.01,1)	0.3672	0.19
$b_{1}$	log-U(0.001,1)	0.0269	0.0120
$b_{2}$	log-U(0.001,1)	0.0119	0.0516
$b_{3}$	log-U(0.001,1)	0.0260	0.0145
FracSevere	log-U(0.01,0.08)	0.0182	0.0253
$FracCritical_{1}$	log-U(0.001,0.02)	0.0041	0.005
$FracCritical_{2}$	log-U(0.001,0.02)	0.0055	0.007
$FracCritical_{3}$	log-U(0.001,0.02)	0.0045	0.0035
$FracCritical_{4}$	log-U(0.001,0.02)	0.0053	0.0048
$FracCritical_{5}$	log-U(0.001,0.02)	0.0033	-
FracAsym	U(0.2,0.7)	0.488	0.4618
IncubPeriod	U(4,6)	4.6389	5.2668
DurMildInf	U(5,30)	15.6157	9.8982
DurAsym	U(5,20)	10.7014	14.5149
DurHosp	U(4,30)	13.2214	15.6204
TimeICUDeath	U(4,30)	11.1608	12.6742
ProbDeath	U(10,90)	31.0621	34.7989
$ProbDeath_{H}$	U(10,90)	22.2085	25.3705
PresymPeriod	log-U(0.5,0.9)	0.6075	0.637
*n*	U(1,100)	47.1727	47.1717
*K*	U(1,$10^{5}$)	$2.48\times 10^{4}$	$4.44\times 10^{4}$
$s_{01}$	log-U(0.4,1.5)	1.0729	1.0409
$s_{02}$	log-U(0.1,0.9)	0.2705	0.3095
$s_{03}$	log-U(0.1,0.9)	0.2815	0.2634
$s_{04}$	log-U(0.4,1.5)	0.432	0.6842
$s_{05}$	log-U(0.4,1.5)	0.5805	0.1837
$s_{06}$	log-U(0.1,0.9)	0.2952	-
$s_{07}$	log-U(0.1,0.9)	0.3077	-

**Table 2 vaccines-09-00915-t002:** Future scenarios simulated for Umbria and Italy through variation of the vaccination rate and of intervention parameters. The three values on the first column represent the increase of transmission parameters on 26 April, 24 May, and 21 June 2021, respectively. The first row indicates the number of first doses of vaccine per day.

		Vaccination Rate					
		Umbria			Italy		
		$8\times 10^{3}$	$12.5\times 10^{3}$	$18\times 10^{3}$	$5\times 10^{5}$	$8\times 10^{5}$	$10^{6}$
**Intervention**	[0.4–0.6–0.8]	Low/Low	Low/Medium	Low/High	Low/Low	Low/Medium	Low/High
	[0.6–0.8–1]	Medium/Low	Medium/Medium	Medium/High	Medium/Low	Medium/Medium	Medium/High
	[0.8–1–1.2]	High/Low	High/Medium	High/High	High/Low	High/Medium	High/High

## Data Availability

Publicly available datasets were analyzed in this study. This data can be found here: https://github.com/pcm-dpc/COVID-19, https://github.com/italia/covid19-opendata-vaccini (accessed on 2 May 2021).

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
