# Peer review of "A Modeling Study on Vaccination and Spread of SARS-CoV-2 Variants in Italy"

_vaccines, 2021, doi:10.3390/vaccines9080915_

Round 1
Reviewer 1 Report
I think, the mathematical model developed and analyzed in this study has fatal flaws. These are as follows:
Several studies that have come out now that show that there are, at least, two major types of protection afforded by COVID-19 vaccines (e.g., mRNA vaccines) to vaccinated persons. One protection is due to vaccine effectiveness (VE) against infection and the other protection is due to VE against disease/symptom. The former VE (i.e., VE against infection) means that fully vaccinated individuals (i.e., individuals who have received two mRNA vaccine doses and 14 days have elapsed since the second dose was injected) are protected from being infected with SARS-CoV-2 (which the authors of this study have considered).
The latter (i.e., VE against disease) means that fully vaccinated individuals are protected against disease if they become infected as a results of breakthrough infection (although rare, breakthrough infection is likely because COVID-19 vaccines are not 100% efficacious). This second VE is not considered by the authors. Please let me know if I got it all wrong.
In addition, the authors do not consider, in the model, vaccinating those individuals who become exposed to SARS-CoV-2 and subsequently either develop COVID-19 symptoms or remain asymptomatic until they recover. Ignoring this aspect of a COVID-19 two-dose vaccine regimen (i.e., not completing a full course) is highly unrealistic. Person vaccinated with the first dose is likely to take their second dose unless that person is severely symptomatic and hospitalized (in which case they are likely to be advised by doctors to delay their second dose).
The authors are strongly encouraged to consider revising their paper considering the above points, which will (at least, I think) improve their paper greatly.
Reviewer 2 Report
This manuscript addresses a very important and timely subject regarding the vaccination and nonpnarmacological intervention (NPI) strategy for the control of Covid-19. The studies reviewed the parameters in Italy and Umbria, where the first outbreak of the SARS-CoV-II variant occurred. The authors used appropriate statistical models, taking account of the dates of vaccination roll-out, and the major NPI events to show that these events combined to have effects on the dynamics of Covid-19 outbreaks. These studies will have significant implication for future management of the Covid outbreaks. Overall, the studies were well done, and the statistical methods used are appropriate. The only caveat is that the major NPI events cited in this study are speculative. There are alternative events which may skew the shape and interpretation of the model. For example, school closure may not be the dominant event that affected the outcome of the shape of the curve of the outbreaks. Alternative events should be considered.
